# Preparation of Gold Nanoparticles via Anodic Stripping of Copper Underpotential Deposition in Bulk Gold Electrodeposition for High-Performance Electrochemical Sensing of Bisphenol A

**DOI:** 10.3390/molecules28248036

**Published:** 2023-12-11

**Authors:** Zhao Huang, Zihan Chen, Dexuan Yan, Shuo Jiang, Libo Nie, Xinman Tu, Xueen Jia, Thomas Wågberg, Long Chao

**Affiliations:** 1Hunan Key Laboratory of Biomedical Nanomaterials and Devices, College of Life Sciences and Chemistry, Hunan University of Technology, Zhuzhou 412007, China; huang1020@hut.edu.cn (Z.H.); chenzh190923@163.com (Z.C.); yandexun0@163.com (D.Y.); shuoj@hunnu.edu.cn (S.J.); libonie@aliyun.com (L.N.); xueen.jia@umu.se (X.J.); 2Key Laboratory of Jiangxi Province for Persistent Pollutants Control and Resources Recycle, Nanchang Hangkong University, Nanchang 330063, China; tuxinman@126.com; 3Department of Physics, Umeå University, SE-901 87 Umeå, Sweden; thomas.wagberg@umu.se

**Keywords:** Au nanoparticles, carbon nanotube, underpotential deposition, electrochemical detection, bisphenol A

## Abstract

Bisphenol A is one of the most widely used industrial compounds. Over the years, it has raised severe concern as a potential hazard to the human endocrine system and the environment. Developing robust and easy-to-use sensors for bisphenol A is important in various areas, such as controlling and monitoring water purification and sewage water systems, food safety monitoring, etc. Here, we report an electrochemical method to fabricate a bisphenol A (BPA) sensor based on a modified Au nanoparticles/multiwalled carbon nanotubes composite electrocatalyst electrode (AuCu-UPD/MWCNTs/GCE). Firstly, the Au-Cu alloy was prepared via a convenient and controllable Cu underpotential/bulk Au co-electrodeposition on a multiwalled modified carbon nanotubes glassy carbon electrode (GCE). Then, the AuCu-UPD/MWCNTs/GCE was obtained via the electrochemical anodic stripping of Cu underpotential deposition (UPD). Our novel prepared sensor enables the high-electrocatalytic and high-performance sensing of BPA. Under optimal conditions, the modified electrode showed a two-segment linear response from 0.01 to 1 µM and 1 to 20 µM with a limit of detection (LOD) of 2.43 nM based on differential pulse voltammetry (DPV). Determination of BPA in real water samples using Au_Cu-UPD_/MWCNTs/GCE yielded satisfactory results. The proposed electrochemical sensor is promising for the development of a simple, low-cost water quality monitoring system for the detection of BPA in ambient water samples.

## 1. Introduction

Bisphenol A (BPA) has been widely used in the production of synthetic polycarbonate and epoxy resin consumer goods due to its good optical transparency, ductility and high heat resistance [1], for example, in a wide range of electronic equipment, building materials, medical equipment, paper products, clothing, plastic bottles, food containers, etc. [2,3,4,5,6]. BPA can be easily released from consumer goods into environmental media and cause human exposure through dietary intake or in other ways [7]. The widespread use of BPA can cause health risks through endocrine-disrupting effects, causing abnormal reproductive development, genotoxicity, and immunotoxicity [8,9,10]. Therefore, the development of highly sensitive, simple, and rapid methods for BPA detection is essential for public health, food, and environmental safety. A variety of analytical techniques for BPA detection have been commonly used, such as high-performance chromatography coupled with mass spectrometry [11], chemiluminescence [12], enzyme-linked immunosorbent assay [13]**,** electroanalysis, etc. [14]. The electroanalysis methods are favorable for rapid BPA detection due to their advantages of low cost, high sensitivity, simplicity, and portability.

Au and its nanocomposites are the most-used electrocatalysts for the electrooxidation and electroanalysis of BPA due to their excellent performance and attractive material characteristics, [4,15,16,17,18,19]. Nevertheless, the large-scale commercial application of Au has been severely limited mainly because of its high price. Therefore, it is significant to reduce the usage and improve the electrocatalytic/electroanalysis performance of gold on the electrode surface. Carbon nanotubes and their composites have been exploited broadly for the electrochemical sensing of BPA because of their low cost, feasibility of modification, and favorable electrochemical properties [14,20,21,22,23]. Generally, the current signal, response speed, and sensitivity will be increased during electroanalysis if the electrochemical reaction rate between the analyte and electrode interface can be substantially improved, which is heavily dependent on the development and innovation of electrocatalysts. We believe that it is valuable to invent a simple and efficient method to prepare Au/MWCNT nanocomposites, which will improve the electrocatalytic performance in BPA detection and reduce the usage of Au.

Owing to special interatomic forces, some metallic atomic monolayers (such as Cu, Pb) can form on the surface of specific noble metals (such as Au, Pt) through underpotential deposition (UPD) instead of bulk electrodeposition (BD), which has been widely studied and applied [24,25]. The structure of metals is regulated at the atomic level using the UPD method, which can change the properties of metallic materials. For example, it is a simple and important strategy to construct sub/monolayer noble metal nanofilms through a galvanic replacement reaction (GRR) between the UPD of a Cu or an Pb metal atom monolayer (reductant) and a noble metal salt (oxidant) [24,26,27]. Recently, a PtAu composite electrocatalyst prepared via Bi-underpotential/PtAu-bulk co-electrodeposition and the subsequent chemical dissolution of Bi for the electrooxidation and electroanalysis of formaldehyde were reported [28]. The BD of noble metal nanoparticles can be dynamically regulated and interfere with the UPD of a less-noble metal at the atomic level via the simultaneous UPD of a less-noble metal and the BD of noble metals. Chemical dissolution of UPD metals in the BD of noble metals is uncontrollable and inconvenient compared with electrochemical anodic stripping, and there are only a few reports on the improvement in noble metals’ electrocatalyst performance via the regulation and interference of UPD metals for advanced electroanalysis studies. Thus, it is significant and attractive to further develop such noble metals and composite electrocatalysts with a special nanostructure prepared via metal underpotential/bulk electrodeposition and a subsequent anodic stripping strategy in the electrochemical analysis field.

Herein, we suggest a new and effective electrochemical method for preparing Au nanocomposite electrocatalysts via Cu underpotential/bulk Au co-electrodeposition on a glassy carbon electrode modified with multiwalled carbon nanotubes, followed by the anodic stripping of Cu UPD. We find that the modified electrode (Au_Cu-UPD_/MWCNTs/GCE) shows excellent electrooxidation activity with BPA, and the high-performance electrochemical sensing of BPA is determined via differential pulse voltammetry (DPV). The modified electrode is satisfactorily applied for the electrochemical sensing of BPA in real water samples.

## 2. Results and Discussion

### 2.1. Fabrication and Characterization of Modified Electrodes

The three steps for Au_Cu-UPD_/MWCNTs/GCE preparation are briefly given below for clarity (with a detailed description in Section 3). The steps are: (1) preparation of MWCNTs/GCE by drop-casting of MWCNTs on the cleaned GCE; (2) Cu UPD and Au BD simultaneously conducted at the potential of Cu UPD on MWCNTs/GCE; (3) preparation of Au_Cu-UPD_/MWCNTs/GCE via anodic stripping for the removal of Cu UPD. For comparison, the conventional Au-modified MWCNTs/GCE (Au_Con_/MWCNTs/GCE) was prepared via conventional electrodeposition at the constant potential, and another Au-modified MWCNTs/GCE (Au_Cu-Con_/MWCNTs/GCE) was prepared via conventional Au-Cu bulk co-electrodeposition at the constant potential, and then the BD Cu was removed via anodic stripping.

First, the electrochemical behavior of Cu^2+^ on the bare Au electrode was investigated using CV and linear sweep anodic stripping voltammetry (LSASV) in 0.2 M aqueous HClO_4_ (as shown in Appendix A, with details discussed in the Appendix A). In order to approximate these experimental conditions, the CV experiment was also conducted on the Au_Con_/MWCNTs/GCE (as shown in Appendix A). The electrochemical behavior of Cu^2+^ on the Au_Con_/MWCNTs/GCE is similar to that of Cu^2+^ on the bare Au electrode according to the CV results. We concluded that the UPD of Cu at 0.1 V can completely avoid the simultaneous BD of Cu on Au and the anodic stripping of Cu UPD can completely finish at 0.6 V. In addition, the UPD of Cu cannot occur on carbon species because the Cu–C interaction is too weak [24,25]. Therefore, only the BD of Au atoms on MWCNTs/GCE occurs, and then the UPD of Cu atoms should happen instantly on the BD of Au here. After a full monolayer of Cu UPD is deposited on the BD Au, the Cu UPD has to stop and only the BD of Au can happen next. Such an alternating process of the BD of Au and then the UPD of Cu continues at the Cu UPD potential for a definite period of time.

Figure 1 shows the cyclic voltammograms of the relevant electrodes in 0.2 M aqueous HClO_4_. No obvious redox peaks were found for the bare GCE. After the modification of MWCNTs on the GCE, we observed a large background current due to the charging on the MWCNTs with high specific surface area. Anodic peaks at potentials higher than ca. 1.0 V were observed for the Au_Con_/MWCNTs/GCE, Au_Cu-Con_/MWCNTs/GCE, and Au_Cu-UPD_/MWCNTs/GCE due to the formation of gold oxides (AuO_x_). Cathodic peaks were observed at ca. 0.9 V, due to the reduction of AuO_x_. Based on a conversion factor of 390 μC cm^−2^, the electrochemical real area (*S*_E_) and the roughness factor (*R*_f_) of Au on the modified electrodes were calculated as reported [29,30]. Briefly, *S*_E_ is the ratio of the cathodic charges of the AuO_x_ reduction peak to the conversion factor (390 μC cm^−2^), and *R*_f_ is the ratio of *S*_E_ to the geometric surface area of the GCE. The cathodic charges under the AuO_x_ reduction peaks follow the order Au_Cu-UPD_/MWCNTs/GCE (366.5 μC) > Au_Cu-Con_/MWCNTs/GCE (74.66 μC) > Au_Con_/MWCNTs/GCE (30.69 μC). Hence, the corresponding *S*_E_ and *R*_f_ values were calculated as follows: *S*_E_ = 0.939 cm^−2^ and *R*_f_ = 13.4 for the Au_Cu-UPD_/MWCNTs/GCE, *S*_E_ = 0.191 cm^−2^ and *R*_f_ = 2.73 for the Au_Cu-Con_/MWCNTs/GCE, and *S*_E_ = 0.0787 cm^−2^ and *R*_f_ = 1.11 for the Au_Con_/MWCNTs/GCE. These findings indicate that the surface area of Au nanoparticles prepared using our protocol is increased (vs. Au _Cu-Con_/MWCNTs/GCE and Au_Con_/MWCNTs/GCE).

The electrodes were also characterized using SEM and EDX, as shown in Figure 2. The bare GCE surface was rather smooth, and the corresponding EDX shows the presence of abundant C (Figure 2A,F). After MWCNTs’ modification, uniformly dispersed and slender carbon nanotubes were observed on the MWCNTs/GCE (Figure 2B). On the Au_Con_/MWCNTs/GCE, many nanoparticles (ca. 80~120 nm) were seen (Figure 2C), and the EDX result confirms the presence of Au (Figure 2G). On the Au_Cu-Con_/MWCNTs/GCE, some smaller nanoparticles (ca. 10~20 nm) and medium-sized nanoparticles (ca. 50~80 nm) were seen (Figure 2E), and the EDX result again confirms the presence of Au (Figure 2H). On the Au_Cu-UPD_/MWCNTs/GCE, many tiny and uniform nanoparticles (ca. 10~20 nm) were seen (Figure 2E), and the EDX result shows more Au load (Figure 2I versus Figure 2G,H) and no signal peak of Cu, which indicate the removal of Cu UPD. Regarding morphology, the SEM results indicate that the Au nanoparticles on the Au_Cu-UPD_/MWCNTs/GCE prepared using our method were smaller and more uniform than those on the Au_Con_/MWCNTs/GCE and the Au_Cu-Con_/MWCNTs/GCE, which could be due to the dynamic interference due to the UPD of Cu resulting in a decrease in Au nanoparticle aggregation during the BD of Au. The smaller Au nanoparticles could increase the specific surface area and roughness, which is consistent with the results of the electrochemical characterization using CV above.

### 2.2. Electrocatalysis and Electrochemical Sensing of BPA

Figure 3 shows the current responses of the relevant electrodes in 0.1 M PBS (pH 7.5) + 10 μM BPA as measured using CV. The oxidation peaks of BPA appeared at ca. 0.48 V~0.50 V during positive scanning, and no obvious peak current appeared during negative scanning. The Au_Cu-UPD_/MWCNTs/GCE shows the highest anodic peak current of 23.9 µA (background current was deducted), which is ca. 6.10, 2.46, 2.22 and 1.64 times that of bare GCE, MWCNTs/GCE Au_Cu-Con_/MWCNTs/GCE, and Au_Con_/MWCNTs/GCE, respectively. The substantial increase in the oxidation peak current on the Au_Cu-UPD_/MWCNTs/GCE may be attributed to the high electrochemically active surface area, the excellent electrocatalytic oxidation capability of Au nanoparticles prepared with the intervention of Cu UPD, and the synergistic catalytic effect on BPA between the smaller gold nanoparticles and MWCNTs. Meanwhile, the capacitive current of the modified electrodes also increased due to the double layer capacitance of the electrode being positively correlated with their surface area.

To better evaluate the electrochemical behavior of BPA, the relationship between the peak current of BPA on the Au_Cu-UPD_/MWCNTs/GCE and the scanning rate was measured using CV in a 0.1 M PBS solution containing 10 µM BPA. As shown in Figure 4A, the oxidation peak current of BPA was gradually increased in the range of 10 to 300 mV s^−1^, and the oxidation potential had a positive shift with increasing scan rate. Figure 4B shows that the peak oxidation current of BPA has a good linear relationship with the scan rate, and the linear regression equation is *I* (μA) = 4.65*v* + 0.437 (*R*^2^ = 0.9901), indicating that the oxidation of BPA on the surface of Au_Cu-UPD_/MWCNTs/GCE was controlled by an adsorption process. Moreover, a plot of the logarithm of peak current versus the logarithm of scan rate is linear, with a slope of 1.12 (as shown in Figure 4C), close to the theoretical value of 1.0 for an adsorption-controlled process. The possible oxidation process of BPA is illustrated in Figure 1 [20,31].

The detection conditions were optimized to obtain the best sensing performance. First, the effect of co-electrodeposition time for preparing the Au_Cu-UPD_/MWCNTs/GCE on the voltammetry signal was examined. As shown in Appendix A, the oxidation peak current of BPA reached a maximum at 800 s according to the CV results, because a too-short co-electrodeposition time cannot enable enough electrodeposition of Au, and too much time would probably result in superabundant electrodeposited Au with the intervention of Cu UPD, which is unfavorable for the formation of smaller Au nanoparticles with a large specific surface area. The pH of the electrolyte has an important influence on the electrochemical behavior of BPA, and, thus, the effect of pH on the oxidation of BPA with the Au_Cu-UPD_/MWCNTs/GCE was studied using CV (as shown in Appendix A). The peak current reached its maximum value at pH = 7.5, so PBS with a pH of 7.5 was selected as the supporting electrolyte here. Finally, the optimization of the preconcentration time using DPV is shown in Appendix A. Increasing the preconcentration time improved the oxidation peak current of BPA. The oxidation peak current reached its maximum value with a preconcentration time of 300 s, which indicates the enrichment of BPA and a saturated signal, letting us select 300 s as the optimum preconcentration time.

Under the optimized conditions described above, the DPV responses and calibration curves of BPA on the Au_Cu-UPD_/MWCNTs/GCE at different concentrations are shown in Figure 5. The oxidation peak current of BPA increases linearly within two concentration ranges of BPA: from 0.01 to 1.0 μM with a sensitivity of 21.6 μA μM^−1^ and from 1.0 to 20 μM with a sensitivity of 1.68 μA μM^−1^, and the linear regression equations are *I* (μA) = 21.6 *c* (μM) + 9.36 (*R*^2^ = 0.9903) and *I* (μA) = 1.68 *c* (μM) + 29.2 (*R*^2^ = 0.9901), respectively (the background currents were deducted here). The limit of detection (LOD) was 2.43 nM (0.555 μg L^−1^). The sensitivity, linear range, and LOD for the electroanalysis of BPA on the Au_Cu-UPD_/MWCNTs/GCE here are better than those of most reported modified electrodes (listed in Table 1), demonstrating the high performance of our developed Au_Cu-UPD_/MWCNTs/GCE BPA sensor.

### 2.3. Anti-Interference Capacity and Stability of the Au_Cu-UPD_/MWCNTs/GCE

In addition, the anti-interference capacity against some interferents was investigated in the presence of 50 nM BPA. Figure 6A,B show that there was very little influence on the DPV response of BPA from the coexistence of 100-fold concentrations of inorganic ions (Zn^2+^, K^+^, Ca^2+^, Pb^2+^, Mg^2+^, Cl^−^, NO^3−^, and SO_4_^2−^), 2, 4-nitrophenol, glucose, and ascorbic acid. In general, the change values of the peak current of BPA are below 5%, showing the good anti-interference capacity of the Au_Cu-UPD_/MWCNTs/GCE sensor.

To evaluate the long-term stability of the Au_Cu-UPD_/MWCNTs/GCE, the detection of 5 μM BPA using an identical modified electrode was repeatedly performed every 1–2 days for 30 days using DPV. As shown in Appendix A, the peak currents of BPA revealed no obvious changes, and the relative standard deviation (RSD) of the peak currents for BPA was 4.2%. The results demonstrate that the Au_Cu-UPD_/MWCNTs/GCE has good long-term stability for the electrochemical sensing of BPA. Moreover, we investigated the reproducibility of five Au_Cu-UPD_/MWCNTs/GCEs prepared under the same conditions (Appendix A). An RSD of 4.9% was obtained, suggesting the good reproducibility of our modified electrode.

### 2.4. Practical Water Sample Analysis

Three practical samples (tap water, bottled water, and local Xiangjiang river water) were used for the detection of BPA using the Au_Cu-UPD_/MWCNTs/GCE. The standard addition method was employed here, and the results are listed in Table 2. From the results, no DPV signal response of BPA was found for any of the samples due to the fact that no BPA or only traces of BPA below the LOD were present in the samples. The value of recovery was in the range of 91.5–104.1%. The results are acceptable and demonstrate that our modified electrode has favorable application potential for the practical electrochemical sensing of BPA.

## 3. Materials and Methods

### 3.1. Instrumentation and Reagents

A CHI660D electrochemical workstation (CH Instrument Co., Shanghai, China) and a conventional three-electrode electrolytic cell were used in all electrochemical experiments. The working electrode was a 3 mm diameter disk glassy carbon electrode (GCE) or its modified electrodes (including MWCNTs/GCE, Au_Con_/MWCNTs/GCE, Au_Cu-con_/MWCNTs/GCE, and Au_Cu-UPD_/MWCNTs/GCE), the reference electrode was a KCl-saturated calomel electrode (SCE), and the counter electrode was a graphite rod. All potentials are reported with respect to SCE. Scanning electron microscopy (SEM) characterizations with energy-dispersive X-ray spectroscopy (EDX) were collected on a TESCAN-MIRA3 LMH field-emission scanning electron microscope.

HAuCl_4_·4H_2_O was purchased from Sinopharm Chemicals Co., Ltd. (Shanghai, China). Bisphenol A, HClO_4_, CuSO_4_·5H_2_O, H_2_SO_4_, and K_4_Fe(CN)_6_·3H_2_O were purchased from Chemicals Company of Tianjin (Tianjin, China). Multiwalled carbon nanotubes with carboxylic treatment (MWCNTs) were purchased from Chengdu Organic Chemicals Co., Ltd. (Chengdu, China) Phosphate-buffered saline (PBS) consisting of 0.1 M KH_2_PO_4_-K_2_HPO_4_ + 0.1 M K_2_SO_4_ was used. All chemicals were of analytical grade or better quality, and all the solutions were prepared using Milli-Q ultrapure water (Millipore, Billerica, MA, USA, >18 MΩ cm). All experiments were performed at room temperature (20–25 °C).

### 3.2. Preparation and Characterization of Modified Electrodes

Cleaning of the GCE was conducted as before [30]. Firstly, the GCE was mechanically polished with alumina powder to a mirror finish. After water rinsing, the polished GCE was ultrasonically treated sequentially in water, ethanol, and water for 5 min each to remove residual alumina powder. The GCE was further treated using cyclic voltammetry (CV, −1.0~1.0 V, 50 mV s^−1^) in 0.2 M aqueous HClO_4_ until reproducible cyclic voltammograms were obtained. After water rinsing, the GCE was characterized using CV (0.1~0.6 V, 50 mV s^−1^) in 0.1 M aqueous Na_2_SO_4_ containing 2.0 mM K_4_Fe(CN)_6_. The peak-to-peak separation of the Fe(CN)_6_^3−^/^4−^ redox peaks was below 75 mV, indicating that the GCE had been well cleaned. The resulting MWCNTs were ultrasonically dispersed into N, N-dimethylformamide, and the concentration of MWCNTs was 2 mg mL^−1^.

The Au_Cu-UPD_/MWCNTs/GCE was fabricated as illustrated in Figure 2 and is described below: (1) an aliquot of 5 μL of MWCNT suspension was drop-casted on the cleaned GCE and dried in air to obtain the MWCNTs/GCE; (2) Cu underpotential/Au bulk co-electrodeposition on the MWCNTs/GCE was performed at a Cu underpotential deposition potential (0.1 V vs. SCE) for 800 s in aqueous 10.0 mM CuSO_4_ + 1.0 mM HAuCl_4_ + 0.2 M HClO_4_ under the N_2_ saturation state; (3) Cu UPD was immediately removed as much as possible via anodic stripping at 0.6 V for 600 s in a blank solution containing 0.2 M HClO_4_; after a quick rinse of the as-prepared modified electrode with ultrapure water, an Au-nanoparticle-modified MWCNTs/GCE (Au_Cu-UPD_/MWCNTs/GCE) was obtained.

For comparison, the conventional Au-modified MWCNTs/GCE (Au_Con_/MWCNTs/GCE) was prepared via conventional electrodeposition at a constant potential of 0.1 V for 800 s in aqueous 1.0 mM HAuCl_4_ + 0.2 M HClO_4_, and another Au-modified MWCNTs/GCE (Au_Cu-Con_/MWCNTs/GCE) was prepared via conventional Au-Cu bulk co-electrodeposition at −0.1 V for 800 s in 10.0 mM CuSO_4_ + 1.0 mM HAuCl_4_ + 0.2 M HClO_4_, and then the BD Cu was removed via anodic stripping at 0.6 V. The above modified electrodes were also used for electrochemical measurements.

### 3.3. Electrochemical Measurement of Bisphenol A

Electrochemical investigation of bisphenol A on the electrodes was carried out using CV. The preconcentration of BPA was performed under solution-stirring conditions; DPV was used for the electroanalysis of BPA immediately when solution-stirring stopped. The response current was recorded as the change in value between the oxidation peak current of bisphenol A and the initial background current.

## 4. Conclusions

In summary, we prepared a new Au_Cu-UPD_/MWCNTs/GCE electrode for the high-performance electrooxidation and electrochemical sensing of bisphenol A. The Au_Cu-UPD_/MWCNTs/GCE was prepared via a simple and controllable Cu underpotential/bulk Au co-electrodeposition and the subsequent electrochemical anodic stripping of Cu UPD. Our new electrochemical sensor shows better analysis performance compared with similar sensors reported previously. Based on the fundamental principles of the electrochemistry of UPD and the anodic stripping of UPD metal, the aggregation through the BD of noble metal nanoparticles can be reduced via the dynamic interference of a less-noble metal at the atomic layer level, and the UPD metal can be effectively removed; thus, the dispersibility and utilization of noble metal electrocatalyst has been considerably improved. The preparation strategy is highly efficient and facile to operate, which can be further extended for the preparation of other noble-metals-based composite nanomaterial and electroanalysis applications in the environmental monitoring of BPA and the detection of other harmful substances.

## Data Availability

Data are contained within the article and Appendix A.

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
