# Peer review of "Preparation of Gold Nanoparticles via Anodic Stripping of Copper Underpotential Deposition in Bulk Gold Electrodeposition for High-Performance Electrochemical Sensing of Bisphenol A"

_molecules, 2023, doi:10.3390/molecules28248036_

Round 1
Reviewer 1 Report
Comments and Suggestions for Authors
Authors developed descriebd the fabrication of a new hybrid GCE+CNT+Au electrode for bisphenol A detection. Only minor changes to the text are needed.
-
R 96: plase control the phrase: “…conducted at Cu underpotential deposition potentia on 96 MWCNTs/GCE”
-
R 262, Materials and methods: “The work-262 ing electrode was a 3 mm diameter disk glassy carbon electrode (GCE) or its modified 263 electrode”. What is the modified electrode? Please specify.
-
R132. “…SE Rt values are calculated as follows”. Authors do not show how the two parameters were calculated but just the final results.
-
Figure 5 and text DPV acronym needs definition.
-
English can be improved
Some phrases are not clear. English can be improved.
Author Response
Comments and Suggestions for Authors:
Authors developed descriebd the fabrication of a new hybrid GCE+CNT+Au electrode for bisphenol A detection. Only minor changes to the text are needed.
Thank you so much for your kind suggestions.
- R 96: plase control the phrase: “…conducted at Cu underpotential deposition potentia on 96 MWCNTs/GCE”
Thank you very much, done so. The phrase expression have been revised in the article.
- R 262, Materials and methods: “The work-262 ing electrode was a 3 mm diameter disk glassy carbon electrode (GCE) or its modified 263 electrode”. What is the modified electrode? Please specify.
Thank you very much, done so. The modified electrodes including MWCNTs/GCE, AuCon/MWCNTs/GCE, AuCu-con/MWCNTs/GCE and AuCu-UPD/MWCNTs/GCE.
- R132. “…SERt values are calculated as follows”. Authors do not show how the two parameters were calculated but just the final results.
Thanks a lot. SE is the ratio of the cathodic charges of the AuOx reduction peak to the conversion factor (390 μC cm−2), and Rf is the ratio of SE to the geometric surface area of GCE. Relevant explanation have been added in the text and marked in red.
- Figure 5 and text DPV acronym needs definition.
Thank you very much. The full name of “DPV” is differential pulse voltammetry, as mentioned in the third paragraph of the introduction when it appears for the first time.
- English can be improved
Thank you very much and we have carefully revised the English for full text.
Reviewer 2 Report
Comments and Suggestions for Authors
The article “Preparation of Au nanoparticles via anodic stripping of underpotential deposition-Cu in bulk electrodeposition-Au for high-performance electrochemical sensing of bisphenol A” reports the electrochemical sensing of BPA using a controllable Cu-underpotential/Au-bulk co-electrodeposition on a glassy carbon electrode modified with multiwalled carbon nanotubes. The manuscript is well written and I recommend for publication in Molecules. Therefore, I have suggestions for improving the work.
1. The key-words must be different from the title to get more visibility for your work.
2. BPA could be found in other several matrices. Authors should add some actual references for this: 10.1016/j.electacta.2023.143164; 10.1016/j.chemosphere.2023.139016; 10.1016/j.chemosphere.2023.137948
3. Section 2.2: Figures should appear after the text (i.e. Fig 3).
4. Authors should explain the capacitive behavior of electrodes in Fig 3 in sensing BPA.
5. What about a peak in ca. 0.15V? It is a strong peak in high scan rates…
6. A plot of log I vs log v should be provided to conclude the mass process. BPA usually adsorbs onto electrode surface.
7. A schematic electrochemical oxidation reaction of BPA should be proposed. Using the scan rate studies it can be proposed.
8. Section 2.3 should be revised. Electrocatalysis of what? Authors don’t understand the concept of electrocatalysis.
9. How many times interferents were tested?
10. Table 2 is not in the section 3.4 (that should be 2.4).
Comments on the Quality of English LanguageThe authors should revise the English of manucript.
Author Response
Comments and Suggestions for Authors
The article “Preparation of Au nanoparticles via anodic stripping of underpotential deposition-Cu in bulk electrodeposition-Au for high-performance electrochemical sensing of bisphenol A” reports the electrochemical sensing of BPA using a controllable Cu-underpotential/Au-bulk co-electrodeposition on a glassy carbon electrode modified with multiwalled carbon nanotubes. The manuscript is well written and I recommend for publication in Molecules. Therefore, I have suggestions for improving the work.
Thank you so much for your kind suggestions.
- The key-words must be different from the title to get more visibility for your work.
Thank you very much. Some keyworks have been appropriately replaced in the revised manuscripts.
- BPA could be found in other several matrices. Authors should add some actual references for this: 10.1016/j.electacta.2023.143164; 10.1016/j.chemosphere.2023.139016; 10.1016/j.chemosphere.2023.137948
Thank you very much. The references have been added.
- Section 2.2: Figures should appear after the text (i.e. Fig 3).
Thanks a lot, done so.
- Authors should explain the capacitive behavior of electrodes in Fig 3 in sensing BPA.
Thank you very much. Compared to smooth bare GCE, the capacitive behavior of the modified electrodes (Fig. 3) may come from their increased surface area, because the double layer capacitance of electrode is positively correlated with their surface area.
- What about a peak in ca. 0.15V? It is a strong peak in high scan rates…
Thanks. A peak in ca. 0.15 V probably come from pre-oxidation current for adsorption process of BPA on the carbon nanotube due to stacking interaction of benzene ring. This phenomenon has also been observed in other literature, such as Ecotox. Environ. Safe (2023) 252:114588.
- A plot of log I vs log v should be provided to conclude the mass process. BPA usually adsorbs onto electrode surface.
Thank you very much for the valuable suggestion. Yes, the electrochemical oxidation of BPA at the electrode surface is not a diffusion-controlled process but an adsorption-controlled process here. We are very sorry for the mistake of incorrect abscissa as “v1/2” in the original Figure 4B and we have revised as “v” for the abscissa of Figure 4B. The peak oxidation current of BPA has a good linear relationship with the scan rate. Moreover, a plot of logarithm of peak current versus logarithm of scan rate is linear with a slope of 1.12 (as shown in Figure 4C), almost equal to the theoretical value of 1.0 for an adsorption-controlled process. The discussion is highlighted in red in the revised manuscripts.
- A schematic electrochemical oxidation reaction of BPA should be proposed. Using the scan rate studies it can be proposed.
Thanks. The schematic electrochemical oxidation reaction of BPA (Scheme 2) have been added in the revised manuscripts.
8.Section 2.3 should be revised. Electrocatalysis of what? Authors don’t understand the concept of electrocatalysis.
Thank you very much. Yes, electrocatalysis of BPA is not discussed in Section 2.3 but actually in Section 2.2, and we are very sorry for the mistake of incorrect title for Section 2.3. The title of Section 2.3 has been revised for “Anti-interference capacity and stability of the AuCu-UPD/MWCNTs/GCE”. In addition, the discussion on electrocatalysis of BPA at AuCu-UPD/MWCNTs/GCE have been appropriately increased in Section 2.2.
- How many times interferents were tested?
Thanks a lot, three parallel tests were performed for each interferent.
- Table 2 is not in the section 3.4 (that should be 2.4).
Thank you very much, done so.
Round 2
Reviewer 2 Report
Comments and Suggestions for Authors
Authors improved the manuscript. Therefore, I recommend the publication in Molecules.